# How Language Models Learn Context-Free Grammars

## Abstract

We design experiments to study *how* generative language models, such as GPT, learn context-free grammars (CFGs) — complex language systems with tree-like structures that encapsulate aspects of human logic, natural languages, and programs. CFGs, comparable in difficulty to pushdown automata, can be ambiguous, usually requiring dynamic programming for rule verification. We create synthetic data to show that pre-trained transformers can learn to generate sentences with near-perfect accuracy and impressive diversity, even for quite challenging CFGs. Crucially, we uncover the *mechanisms* behind transformers learning such CFGs. We find that the hidden states implicitly encode the CFG structure (such as putting tree node info exactly on the subtree boundary), and that the transformer can form "boundary to boundary" attentions that mimic dynamic programming. We also discuss CFG extensions and transformer robustness against grammar errors.

## 1 Introduction

Language models (OpenAI, 2023) are neural networks designed to learn the probability distribution of natural language and generate text. Models like GPT (Radford et al., 2018) can accurately follow language structures (Shen et al., 2017; Tenney et al., 2019), even in smaller models (Black et al., 2021). However, the mechanisms and representations these models use to capture language rules and patterns remain unclear. Despite recent theoretical advances in understanding language models (Bhattamishra et al., 2020; Jelassi et al., 2022; Li et al., 2023; Liu et al., 2022; Yao et al., 2021), most are limited to simple settings and fail to account for the complex structure of languages.

In this paper, we explore the **mechanisms** behind generative language models learning probabilistic context-free grammars (CFGs) (Lee, 1996). CFGs, capable of generating a diverse set of *highly structured* expressions, consist of terminal (T) and nonterminal (NT) symbols, a root symbol, and production rules. A string belongs to the language generated by a CFG if there is a sequence of rules that transform the root symbol into the string of T symbols. For instance, the CFG below generates the language of balanced parentheses:

$$s \rightarrow ss \mid (s) \mid \varnothing$$

where $\varnothing$ denotes the empty string. Examples in the language include $\varnothing$, ( ) , ( ( ) ) , ( ) ( ) , ( ( ( ) ) ) .

Many structures in languages can be viewed as CFGs, including *grammars, structures of the codes, mathematical expressions, music patterns, article formats* (for poems, instructions, legal documents), etc. We use transformer (Vaswani et al., 2017) as the generative language model and study how it learns the CFGs. Transformers can encode some CFGs, especially those that correspond to the grammar of natural languages (Arps et al., 2022; Hewitt & Manning, 2019; Manning et al., 2020; Maudslay & Cotterell, 2021; Shi et al., 2022; Vilares et al., 2020; Wu et al., 2020; Zhao et al., 2023). However, the *mechanism* behind how such CFGs can be efficiently learned by transformers remains unclear. Previous works (Deletang et al., 2023) studied transformer's learnability on a few languages in the Chomsky hierarchy (which includes CFGs) but the inner mechanisms regarding how transformer can or cannot solve those tasks remain unclear.

For a generative language model to learn a long CFG (e.g. *hundreds of tokens*), it needs to **efficiently learn many non-trivial, long-distance planning**. The model cannot just generate tokens that are "locally consistent." For example, to generate a string with balanced parentheses, the model must keep track of the number and type of open and close parentheses *globally*. Imagine, for complex CFGs, even verifying that a sequence satisfies a given CFG may require dynamic programming: to have a memory and a mechanism to access the memory in order to verify the hierarchical structure of the CFG. Learning CFGs is thus a significant challenge for the transformer model, and it tests the model's ability to learn and generate complex and diverse expressions.

Figure 1: An example string $x$ from $\mathcal{G} = \mathsf{cfg3f}$. Though formally defined in Section 2, bold symbols in color represent *NT boundaries* which marks the ending positions of the parsed CFG subtrees at various levels $\ell$: we denote by $\mathfrak{b}_\ell(i) = 1$ if position $i$ is at the NT boundary for level $\ell$. The *NT ancestor* $\mathfrak{s}_\ell(i)$ represents the tree node's name at level $\ell$ for a symbol at position $i$.

---

**Remark.** In this paper, we analyze the transformer's ability to learn highly ambiguous CFGs. Even if the CFG rules are given, typically one uses dynamic programming (DP) to decide if $x \in L(\mathcal{G})$.

In this study, we pre-train GPT-2 (Radford et al., 2019) on a language modeling task using a large corpus of strings sampled from a few very non-trivial CFGs that we construct with different levels of difficulties — see Figure 1 for an example and Figure 9 in the appendix for more. We test the model's accuracy and *diversity* by feeding it *prefixes* from the CFG and observing if it can generate accurate completions.

- We show the model can achieve near-perfect CFG generation accuracies.
- We check the model's output distribution / diversity show it is close to that of the true CFG.

Our paper's **key contribution** is an analysis of *how transformers recover the structures of the underlying CFG*, examining attention patterns and hidden states. Specifically, we:

- Develop a probing method to verify that the model's hidden states linearly encode NT information almost perfectly, a significant finding as pre-training does not expose the CFG structure.
- Introduce methods to visualize and quantify attention patterns, demonstrating that GPT learns position-based and boundary-based attentions, contributing to understanding the CFG's regularity, periodicity, and hierarchical structure.
- Suggest that GPT models learn CFGs by *implementing a dynamic programming-like algorithm*. We find that boundary-based attention allows a token to attend to its closest NT symbols in the CFG tree, even when separated by hundreds of tokens. This resembles dynamic programming, in which the CFG parsing on a sequence $1...i$ needs to be "concatenated" with another sequence $i + 1...j$ in order to form a solution to a larger problem on $1...j$. See Figure 1 for an illustration.

We also explore *implicit CFGs* (Post & Bergsma, 2013), where each T symbol is a bag of tokens, and data is generated by randomly sampling tokens. This allows capturing additional structure, like word categories. We demonstrate that the model learns implicit CFGs by encoding the T symbol information in its token embedding layer. We also investigate *model robustness* using CFGs, testing the model's ability to correct errors and generate valid CFGs from a corrupted prefix.

## 2 Context-Free Grammars

A probabilistic context-free grammar (CFG) is a formal system defining a string distribution using production rules. It comprises four components: terminal symbols ($\mathbf{T}$), nonterminal symbols ($\mathbf{NT}$), a root symbol ($root \in \mathbf{NT}$), and production rules ($\mathcal{R}$). We represent a CFG as $\mathcal{G} = (\mathbf{T}, \mathbf{NT}, \mathcal{R})$, with $L(\mathcal{G})$ denoting the string distribution generated by $\mathcal{G}$.

We mostly focus on $L$-level CFGs where each level $\ell \in [L]$ corresponds to a set of symbols $\mathbf{NT}_\ell$ with $\mathbf{NT}_\ell \subseteq \mathbf{NT}$ for $\ell < L$, $\mathbf{NT}_L = \mathbf{T}$, and $\mathbf{NT}_1 = \{root\}$. Symbols at different levels are disjoint: $\mathbf{NT}_i \cap \mathbf{NT}_j = \varnothing$ for $i \neq j$. We consider rules of length 2 or 3, denoted as $\mathcal{R} = (\mathcal{R}_1, \dots, \mathcal{R}_{L-1})$, where each $\mathcal{R}_\ell$ consists of rules in the form:

$$r = (a \mapsto b, c, d) \quad \text{or} \quad r = (a \mapsto b, c) \quad \text{for} \quad a \in \mathbf{NT}_\ell \quad \text{and} \quad b, c, d \in \mathbf{NT}_{\ell+1}$$

Given a non-terminal symbol $a \in \mathbf{NT}$ and any rule $r = (a \mapsto \star)$, we say $a \in r$. For each $a \in \mathbf{NT}$, its associated set of rules is $\mathcal{R}(a) := \{r \mid r \in \mathcal{R}_\ell \wedge a \in r\}$, its *degree* is $|\mathcal{R}(a)|$, and the CFG's *size* is $(|\mathbf{NT}_1|, |\mathbf{NT}_2|, \dots, |\mathbf{NT}_L|)$.

**Generating from CFG.** To generate samples $x$ from $L(\mathcal{G})$, follow these steps:

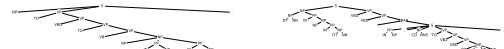

(a) real-life English CFG derived from Penn Treebank, short and simple

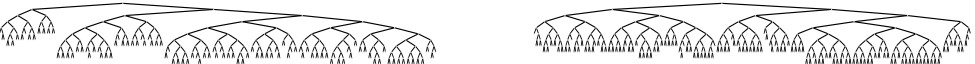

(b) a family of max-depth 11 CFGs where rules have length 1 or 2 that GPT can learn, see cfg0 in Appendix H

Figure 2: CFG visual comparisons: *left* is a medium-length sample, and *right* is a 80%-percentile-length sample

1. Start with the *root* symbol $\mathbf{NT}_1$.
2. For each layer $\ell < L$, keep a sequence of symbols $s_\ell = \big(s_{\ell,1}, \cdots, s_{\ell,m_\ell}\big)$.
3. For the next layer, randomly sample a rule $r \in \mathcal{R}(s_{\ell,i})$ for each $s_{\ell,i}$ with uniform probability.[1] Replace $s_{\ell,i}$ with $b, c, d$ if $r = (s_{\ell,i} \mapsto b, c, d)$, or with $b, c$ if $r = (s_{\ell,i} \mapsto b, c)$. Let the resulting sequence be $s_\ell = \big(s_{\ell+1,1}, \cdots, s_{\ell+1,m_{\ell+1}}\big)$.
4. During generation, when a rule $s_{\ell,i} \mapsto s_{\ell+1,j}, s_{\ell+1,j+1}$ is applied, define the parent $\mathsf{par}_{\ell+1}(j) = \mathsf{par}_{\ell+1}(j+1) := i$ (and similarly if the rule of $s_{\ell,i}$ is of length 3).
5. Define **NT ancestor indices** $\mathfrak{p} = (\mathfrak{p}_1(i), \ldots, \mathfrak{p}_L(i))_{i \in [m_L]}$ and **NT ancestor symbols** $\mathfrak{s} = (\mathfrak{s}_1(i), \ldots, \mathfrak{s}_L(i))_{i \in [m_L]}$ as shown in Figure 1:

$$\mathfrak{p}_L(j) := j \ , \quad \mathfrak{p}_\ell(j) := \mathsf{par}_{\ell+1}(\mathfrak{p}_{\ell+1}(j)) \quad \text{and} \quad \mathfrak{s}_\ell(j) := s_{\ell,\mathfrak{p}_\ell(j)}$$

The final string is $x = s_L = (s_{L,1}, \cdots, s_{L,m_L})$ with $x_i = s_{L,i}$ and length $\mathbf{len}(x) = m_L$. We use $(x, \mathfrak{p}, \mathfrak{s}) \sim L(\mathcal{G})$ to represent $x$ with its associated NT ancestor indices and symbols, sampled according to the generation process. We write $x \sim L(\mathcal{G})$ when $\mathfrak{p}$ and $\mathfrak{s}$ are evident from the context.

**Definition 2.1.** *A symbol $x_i$ in a sample $(x, \mathfrak{p}, \mathfrak{s}) \sim L(\mathcal{G})$ is the **NT boundary / NT end** at level $\ell \in [L-1]$ if $\mathfrak{p}_\ell(i) \neq \mathfrak{p}_\ell(i+1)$ or $i = \mathbf{len}(x)$. We denote $\mathfrak{b}_\ell(i) := \mathbb{1}_{x_i \text{ is the NT boundary at level } \ell}$ as the **NT-end boundary** indicator function. The **deepest NT-end** of $i$ is*

$$\mathfrak{b}^\sharp(i) = \min_{\ell \in \{2,3,\ldots,L-1\}} \{\mathfrak{b}_\ell(i) = 1\} \quad \text{or } \perp \text{ if the set is empty } .$$

**The cfg3 synthetic CFG family.** We focus on seven synthetic CFGs of depth $L = 7$ detailed in Section B.1. The hard datasets cfg3b, cfg3i, cfg3h, cfg3g, cfg3f have sizes $(1, 3, 3, 3, 3, 3, 3)$ and increasing difficulties cfg3b $<$ cfg3i $<$ cfg3h $<$ cfg3g $<$ cfg3f. The easy datasets cfg3e1 and cfg3e2 have sizes $(1, 3, 9, 27, 81, 27, 9)$ and $(1, 3, 9, 27, 27, 9, 4)$ respectively. The sequences generated by these CFGs are up to $3^6 = 729$ in length. Typically, the learning difficulty of CFGs *inversely scales* with the number of NT/T symbols or CFG rules, assuming other factors remain constant (see Figure 3 and more in Appendix H). We thus primarily focus on cfg3b, cfg3i, cfg3h, cfg3g, cfg3f.

**Why Such CFGs.** In this paper, we use CFG as a proxy to study some rich, recursive structure in languages, which can cover some logics, grammars, formats, expressions, patterns, etc. Those structures are diverse yet strict (for example, Section 3.1 should be only followed by Section 3.1.1, Section 4 or Section 3.2, not others). We create a synthetic CFG to approximate such richness and structure. The CFGs we consider are non-trivial, with likely over $2^{270} > 10^{80}$ strings in cfg3f among a total of over $3^{300} > 10^{140}$ possible strings of length 300 or more (see the entropy estimation in Figure 3). The probability of a random string belonging to this language is nearly zero, and a random completion of a valid prefix is unlikely to satisfy the CFG.

Moreover, to probe the *inner workings* of the transformer, we choose a CFG family with a "canonical representation" and show a high correlation between this representation and the hidden states in the learned transformer. Such a *controlled experiment* allows us to better understand the learning process. We also construct additional CFG families to study "not-so-canonical" CFG trees, with results deferred to Appendix H. We do not claim our result captures all CFGs, however, we view our work as a promising starting point: our CFG is already quite challenging for a transformer to learn — for example, in Appendix H, we show that a CFG derived from English Penn TreeBank can be learned well using small models (like GPTs with $\sim$ 100k parameters), whereas our cfg3 family requires GPT2 with 100M parameters — yet we can still identify how transformer learns it.

---

[1]For simplicity, we consider the uniform case, eliminating rules with extremely low probability. Such rules complicate the learning of the CFG and the investigation of a transformer's inner workings. Our results can easily extend to non-uniform cases, provided the distributions are not heavily unbalanced.

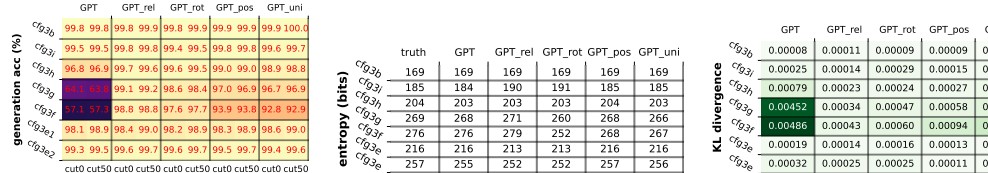

Figure 3: Generation accuracy (left), entropy (middle), KL-divergence (right) across multiple CFG datasets. **Observation:** Less ambiguous CFGs (cfg3e1, cfg3e2, as they have fewer NT/T symbols) are easier to learn. Modern transformer variants using relative positional embedding ($\mathrm{GPT_{rel}}$ or $\mathrm{GPT_{pos}}$) are better for learning complex CFGs. We also present weaker variants $\mathrm{GPT_{pos}}$ and $\mathrm{GPT_{uni}}$ that base their attention matrices solely on token positions (serving specific purposes in Section 5.1).

## 3 TRANSFORMER LEARNS SUCH CFGS

In this section, we evaluate the generative capability of the transformer by testing its accuracy in completing sequences from prefixes of strings in $L(\mathcal{G})$. We also evaluate the diversity of the generated outputs and verify if the distribution of these strings aligns with the ground truth $L(\mathcal{G})$.

**Models.** We denote the vanilla GPT2 small architecture (12-layer, 12-head, 768-dimensions) as GPT (Radford et al., 2019). Given GPT2's weak performance due to its absolute positional embedding, we implemented two modern variants. We denote GPT with relative positional attention (He et al., 2020) as $\mathrm{GPT_{rel}}$, and GPT with rotary positional embedding (Black et al., 2022; Su et al., 2021) as $\mathrm{GPT_{rot}}$. For specific purposes in later sections, we introduce two weaker variants of GPT. $\mathrm{GPT_{pos}}$ replaces the attention matrix with a matrix based solely on tokens' relative positions, while $\mathrm{GPT_{uni}}$ uses a constant, uniform average of past tokens from various window lengths as the attention matrix. Detailed explanations of these variants are in Section B.2.

**Completion accuracy.** We generate a large corpus $\{x^{(i)}\}_{i\in[N]}$ from a synthetic CFG $\mathcal{G}$ as described in Section 2. A model $F$ is pretrained on this corpus, treating each terminal symbol as a separate token, using an auto-regressive task (Section B.3 for details). For evaluation, $F$ generates completions for prefixes $x_{:c} = (x_1, x_2, \cdots, x_c)$ from strings $x$ freshly generated from $L(\mathcal{G})$. The *generation accuracy* is measured as $\mathbf{Pr}_{x \sim L(G) + \text{randomness of } F}[(x_{:c}, F(x_{:c})) \in L(\mathcal{G})]$. We use multinomial sampling without beam search for generation.[2]

Figure 3 (left) shows the generation accuracies for cuts $c = 0$ and $c = 50$. The $c = 0$ result tests the transformer's ability to generate a sentence in the CFG, while $c = 50$ tests its ability to complete a sentence.[3] The results show that the pretrained transformers can generate near-perfect strings that adhere to the CFG rules for the cfg3 data family.

**Generation diversity.** Could it be possible that the trained transformer only memorized a small subset of strings from the CFG? We evaluate its learning capability by measuring the diversity of its generated strings. High diversity suggests a better understanding of the CFG rules.

Diversity can be estimated through entropy. Given a distribution $p$ over strings and a sampled subset $S = \{x^{(i)}\}_{i\in[M]}$ from $p$, for any string $x \in S$, denote by $\mathbf{len}(x)$ its length so $x = (x_1, \ldots, x_{\mathbf{len}(x)})$, and denote by $x_{\mathbf{len}(x)+1} = \mathsf{eos}$. The entropy in bits for $p$ can be estimated by

$$-\tfrac{1}{|S|}\sum_{x\in S}\sum_{i\in[\mathbf{len}(x)+1]}\log_2 \mathbf{Pr}_p\left[x_i \mid x_1, \ldots, x_{i-1}\right]$$

We compare the entropy of the true CFG distribution and the transformer's output distribution using $M = 20000$ samples in Figure 3 (middle).

Diversity can also be estimated using the birthday paradox to lower bound the support size of a distribution (Arora & Zhang, 2017). Given a distribution $p$ over strings and a sampled subset $S = \{x^{(i)}\}_{i\in[M]}$ from $p$, if every pair of samples in $S$ are distinct, then with good probability the support of $p$ is of size at least $\Omega(M^2)$. In Appendix C.1, we conducted an experiment with $M = 20000$. We performed a birthday paradox experiment from every symbol $a \in \mathbf{NT}_{\ell_1}$ to some other level

---

[2]The last softmax layer converts the model outputs into a probability distribution over (next) symbols. We follow this distribution to generate the next symbol, reflecting the unaltered distribution learned by the transformer. This is the source of the "randomness of $F$" and is often referred to as using "temperature $\tau = 1$."

[3]Our cfg3 family is large enough to ensure a negligible chance of a freshly sampled prefix of length 50 being seen during pretraining.

$\ell_2 > \ell_1$, comparing that with the ground truth. For instance, we confirmed for the cfg3f dataset, there are at least $\Omega(M^2)$ distinct sequences to level 5 generated from a symbol $a \in \mathbf{NT}_2$ — not to mention from the root in $\mathbf{NT}_1$ to the leaf at level 7. In particular, $M^2$ is already more than the number of parameters in the model. From both experiments, we conclude that the pre-trained model **does not rely on simply memorizing** a small set of patterns to learn the CFGs.

**Distribution comparison.** To fully learn a CFG, it is crucial to learn the distribution of generating probabilities. However, comparing distributions of exponential support size can be challenging. A naive approach is to compare the marginal distributions $p(a, i)$, which represent the probability of symbol $a \in \mathbf{NT}_\ell$ appearing at position $i$ (i.e., the probability that $\mathfrak{s}_\ell(i) = a$). We observe a strong alignment between the generation probabilities and the ground-truth distribution, see Appendix C.2.

Another approach is to compute the KL-divergence between the per-symbol conditional distributions. Let $p^*$ be the distribution over strings in the true CFG and $p$ be that from the transformer model. Let $S = \left\{ x^{(i)} \right\}_{i \in [M]}$ be samples from the true CFG distribution. Then, the KL-divergence can be estimated as follows:[4]

$$\frac{1}{|S|} \sum_{x \in S} \frac{1}{\mathbf{len}(x)+1} \sum_{i \in [\mathbf{len}(x)+1]} \sum_{t \in \mathbf{T} \cup \{\mathbf{eos}\}} \mathbf{Pr}_{p^*}[t \mid x_1, \ldots, x_{i-1}] \log \frac{\mathbf{Pr}_{p^*}[t|x_1,\ldots,x_{i-1}]}{\mathbf{Pr}_p[t|x_1,\ldots,x_{i-1}]}$$

In Figure 3 (right) we compare the KL-divergence between the true CFG distribution and the transformer's output distribution using $M = 20000$ samples.

# 4 HOW DO TRANSFORMERS LEARN CFGS?

In this section, we delve into the learned representation of the transformer to understand *how* it encodes CFGs. We employ various measurements to probe the representation and gain insights.

**Recall classical way to solve CFGs.** Given CFG $\mathcal{G}$, the classical way to verify if a sequence $x$ satisfies $L(\mathcal{G})$ is to use dynamic programming (DP) (Sakai, 1961; Sipser, 2012). One possible implementation of DP involves using the function $\mathsf{DP}(i, j, a)$, which determines whether or not $x_i, x_{i+1} \ldots, x_j$ can be generated from symbol $a$ following the CFG rules. From this DP representation, a DP recurrent formula can be easily derived.[5]

In the context of this paper, any sequence $x \sim L(\mathcal{G})$ that satisfies the CFG must satisfy the following conditions (recall the NT-boundary $\mathfrak{b}_\ell$ and the NT-ancestor $\mathfrak{s}_\ell$ notions from Section 2):

$$\mathfrak{b}_\ell(i-1) = 1, \mathfrak{b}_\ell(j) = 1, \forall k \in [i,j), \mathfrak{b}_\ell(k) = 0 \text{ and } \mathfrak{s}_\ell(i) = a \implies \mathsf{DP}(i,j,a) = 1 \quad (4.1)$$

Note that (4.1) is not an "if and only if" condition because there may be a subproblem $\mathsf{DP}(i, j, a) = 1$ that does not lie on the final CFG parsing tree but is still locally parsable by some valid CFG subtree. However, (4.1) provides a "backbone" of subproblems, where verifying their $\mathsf{DP}(i, j, a) = 1$ values *certifies* that the sentence $x$ is a valid string from $L(\mathcal{G})$. It is worth mentioning that ***depending on the implementation of a DP program*** (e.g., different orders on pruning or binarization), ***not all*** $(i, j, a)$ tuples need to be computed in $\mathsf{DP}(i, j, a)$. Only those in the "backbone" are necessary.

**Connecting to transformer.** In this section, we investigate whether pre-trained transformer $F$ not only generates grammatically correct sequences, but also implicitly encodes the NT ancestor and boundary information. If it does, this suggests that the transformer contains sufficient information to support all the $\mathsf{DP}(i, j, a)$ values in the backbone. This is a significant finding, considering that transformer $F$ is trained solely on the auto-regressive task without any exposure to NT information. If it does encode the NT information after pretraining, it means that the model can both generate and certify sentences in the CFG language.

## 4.1 FINDING 1: TRANSFORMER'S HIDDEN STATES ENCODE NT ANCESTORS AND BOUNDARIES

Let $l$ be the *last layer* of the transformer (other layers are considered in Appendix D.2). Given an input string $x$, the hidden state of the transformer at layer $l$ and position $i$ is denoted as

---

[4]A nearly identical formula was also used in DuSell & Chiang (2022).

[5]For example, one can compute $\mathsf{DP}(i, j, a) = 1$ if and only if there exists $i = i_1 < i_2 < \cdots < i_k = j + 1$ such that $\mathsf{DP}(i_r, i_{r+1}-1, b_r) = 1$ for all $r \in [k-1]$ and $a \to b_1, b_2, \ldots, b_k$ is a rule of the CFG. Implementing this naively would result in a $O(\mathbf{len}^4)$ algorithm for CFGs with a maximum rule length of 3. However, it can be implemented more efficiently with $O(\mathbf{len}^3)$ time by introducing auxiliary nodes (e.g., via binarization).

| | GPT | | | | | GPT_rel | | | | | GPT_rot | | | | | GPT_pos | | | | | GPT_uni | | | | | deBERTa | | | | | baseline (GPT_rand) | | | | |
|---|---|---|---|---|---|---|---|---|---|---|---|---|---|---|---|---|---|---|---|---|---|---|---|---|---|---|---|---|---|---|---|---|---|---|---|
| $cfg_{3b}$ | 100 | 100 | 100 | 100 | 100 | 100 | 100 | 100 | 100 | 100 | 100 | 100 | 100 | 100 | 100 | 100 | 100 | 100 | 100 | 100 | 100 | 100 | 100 | 100 | 100 | 100 | 100 | 100 | 99.7 | 99.9 | 85.0 | 65.7 | 56.8 | 61.5 | 62.7 |
| $cfg_{3i}$ | 99.6 | 99.7 | 99.6 | 99.2 | 99.7 | 99.6 | 99.7 | 99.6 | 99.2 | 99.7 | 99.6 | 99.7 | 99.6 | 99.2 | 99.8 | 99.6 | 99.7 | 99.6 | 99.3 | 99.8 | 99.6 | 99.7 | 99.6 | 99.3 | 99.8 | 99.7 | 99.7 | 99.7 | 99.2 | 99.4 | 84.6 | 71.7 | 64.6 | 66.4 | 65.2 |
| $cfg_{3h}$ | 99.7 | 98.3 | 98.3 | 99.2 | 100 | 99.7 | 98.1 | 97.8 | 99.0 | 100 | 99.7 | 98.4 | 98.2 | 99.3 | 100 | 99.7 | 98.5 | 98.5 | 99.4 | 100 | 99.7 | 98.6 | 98.6 | 99.4 | 100 | 99.9 | 99.8 | 98.9 | 99.7 | 100 | 67.5 | 47.2 | 50.6 | 66.3 | 92.8 |
| $cfg_{3g}$ | 100 | 99.2 | 95.6 | 94.6 | 97.3 | 100 | 99.3 | 96.7 | 97.2 | 99.0 | 100 | 99.3 | 96.6 | 97.2 | 99.0 | 100 | 99.3 | 96.7 | 96.9 | 98.8 | 100 | 99.4 | 97.0 | 97.2 | 98.9 | 100 | 99.5 | 95.5 | 85.6 | 90.5 | 70.8 | 56.4 | 49.4 | 57.0 | 73.1 |
| $cfg_{3f}$ | 100 | 97.6 | 94.3 | 88.4 | 85.9 | 100 | 97.5 | 94.8 | 92.9 | 93.5 | 100 | 97.7 | 95.2 | 93.3 | 94.2 | 100 | 97.9 | 95.6 | 93.5 | 93.9 | 100 | 98.2 | 95.8 | 93.2 | 93.5 | 100 | 99.6 | 96.3 | 84.0 | 77.5 | 71.3 | 49.9 | 44.6 | 59.1 | 68.6 |
| $cfg_{3e1}$ | 100 | 100 | 100 | 100 | 100 | 100 | 100 | 100 | 100 | 100 | 100 | 100 | 100 | 100 | 100 | 100 | 100 | 100 | 100 | 100 | 100 | 100 | 100 | 100 | 100 | 100 | 100 | 100 | 100 | 99.8 | 45.4 | 27.6 | 34.6 | 47.2 | 76.3 |
| $cfg_{3e2}$ | 99.9 | 100 | 100 | 100 | 100 | 99.8 | 100 | 100 | 100 | 100 | 99.9 | 100 | 100 | 100 | 100 | 99.9 | 100 | 100 | 100 | 100 | 99.9 | 100 | 100 | 100 | 100 | 100 | 100 | 100 | 100 | 99.9 | 36.0 | 16.6 | 23.5 | 44.6 | 78.3 |
| | NT6 | NT5 | NT4 | NT3 | NT2 | NT6 | NT5 | NT4 | NT3 | NT2 | NT6 | NT5 | NT4 | NT3 | NT2 | NT6 | NT5 | NT4 | NT3 | NT2 | NT6 | NT5 | NT4 | NT3 | NT2 | NT6 | NT5 | NT4 | NT3 | NT2 | NT6 | NT5 | NT4 | NT3 | NT2 |

*(y-axis: predict NT ancestor (%))*

Figure 4: After pre-training, hidden states of generative models implicitly encode the NT ancestors information. The $NT_\ell$ column represents the accuracy of predicting $\mathfrak{s}_\ell$, the NT ancestors at level $\ell$.

It also encodes NT boundaries, see Appendix D.1; and such information is discovered gradually and *hierarchically*, across layers and training epochs, see Appendix D.2 and D.3. We compare against a baseline which is the encoding from a random GPT. We also compare against DeBERTa, illustrating that BERT-like models are less effective in learning NT information at levels close to the CFG root.

$E_i(x) \in \mathbb{R}^d$. We investigate whether a linear function can predict $\big(\mathfrak{b}_1(i), \ldots, \mathfrak{b}_L(i)\big)_{i \in [\mathbf{len}(x)]}$ and $\big(\mathfrak{s}_1(i), \ldots, \mathfrak{s}_L(i)\big)_{i \in [\mathbf{len}(x)]}$ using only $\big(E_i(x)\big)_{i \in [\mathbf{len}(x)]}$. If possible, it implies that the last-layer hidden states *encode the CFG's structural information up to a linear transformation*.

**Our multi-head linear function.** Due to the high dimensionality of this linear function (e.g., $\mathbf{len}(x) = 300$ and $d = 768$ yield $300 \times 768$ dimensions) and *variable string lengths*, we propose a multi-head linear function for efficient learning. We consider a set of linear functions $f_r \colon \mathbb{R}^d \to \mathbb{R}^{|\mathbf{NT}|}$, where $r \in [H]$ and $H$ is the number of "heads". To predict any $\mathfrak{s}_\ell(i)$, we apply:

$$G_i(x) = \sum_{r \in [H], k \in [\mathbf{len}(x)]} w_{r,i \to k} \cdot f_r(E_k(x)) \in \mathbb{R}^{|\mathbf{NT}|} \tag{4.2}$$

where $w_{r,i \to k} := \frac{\exp(\langle P_{i,r}, P_{k,r} \rangle)}{\sum_{k' \in [\mathbf{len}(x)]} \exp(\langle P_{i,r}, P_{k',r} \rangle)}$ for trainable parameters $P_{i,r} \in \mathbb{R}^{d'}$. $G_i$ can be seen as a "multi-head attention" over linear functions. We train $G_i(x) \in \mathbb{R}^{|\mathbf{NT}|}$ using the cross-entropy loss to predict $\big(\mathfrak{s}_\ell(i)\big)_{\ell \in [L]}$. Despite having multiple heads,

$$G_i(x) \text{ is still a linear function over } \big(E_k(x)\big)_{k \in [\mathbf{len}(x)]}$$

as the linear weights $w_{r,i \to k}$ depend only on positions $i$ and $k$, not on $x$. Similarly, we train $G'_i(x) \in \mathbb{R}^L$ using the logistic loss to predict the values $\big(\mathfrak{b}_\ell(i)\big)_{\ell \in [L]}$. Details are in Section B.4.

**Results.** Our experiments (Figure 4) suggest that pre-training allows the generative models to *almost perfectly encode* the NT ancestor and NT boundary information in the last transformer layer's hidden states, up to a *linear* transformation.

## 4.2 FINDING 2: TRANSFORMER'S HIDDEN STATES ENCODE NT ANCESTORS AT NT BOUNDARIES

We previously used the *entire* hidden state layer, $\big(E_i(x)\big)_{i \in [\mathbf{len}(x)]}$, to predict $\big(\mathfrak{s}_\ell(i)\big)_{\ell \in [L]}$ for *each* position $i$. This is essential for a generative/decoder model as it's impossible to extract $i$'s NT ancestors by only examining $E_i(x)$ or the hidden states to its *left*, especially if a token $x_i$ is near the string's start or a subtree's starting token in the CFG.

However, if we only consider a neighborhood of position $i$ in the hidden states, say $E_{i\pm 1}(x)$, what can we infer from it through linear probing? We can replace $w_{r,i \to k}$ in (4.2) with a replace $w_{r,i \to k}$ with zeros for $|i - k| > 1$ (tridiagonal masking), or with zeros for $i \neq k$ (diagonal masking).

**Results.** We observe two key points. First, diagonal or tridiagonal masking is sufficient for predicting NT boundaries, i.e., $\mathfrak{b}_\ell(i)$, with decent accuracy (deferred to Figure 15 in Appendix D.1). More importantly, at NT boundaries (i.e., $\mathfrak{b}_\ell(x) = 1$), such masking is adequate for accurately predicting the NT ancestors $\mathfrak{s}_\ell(x)$ (see Figure 5). Hence, we conclude that the information of position $i$'s NT ancestors is *locally encoded around position $i$ when $i$ is on the NT boundary*.

**Related work.** Our probing approach is akin to the seminal work by Hewitt & Manning (2019), which uses linear probing to examine the correlation between BERT's hidden states and the parse tree distance metric (similar to NT-distance in our language). Subsequent studies (Arps et al., 2022; Manning et al., 2020; Maudslay & Cotterell, 2021; Shi et al., 2022; Vilares et al., 2020; Wu et al., 2020; Zhao et al., 2023) have explored various probing techniques to suggest that BERT-like transformers can approximate CFGs from natural languages.

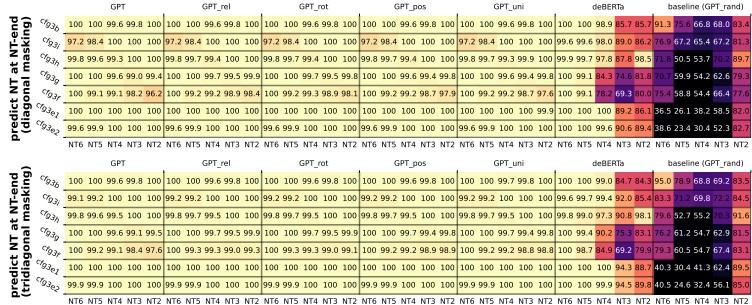

**Observation.** BERT-like (encoder-only) transformers, such as De-BERTa, trained on a masked language modeling (MLM) task, do not store deep NT ancestor information at the NT boundaries.

Figure 5: Generative pre-trained transformer encodes NT ancestors almost exactly at NT boundaries. The $NT_\ell$ column represents the linear-probing accuracy of predicting $\mathfrak{s}_\ell(i)$ at locations $i$ with $\mathfrak{b}_\ell(i) = 1$.

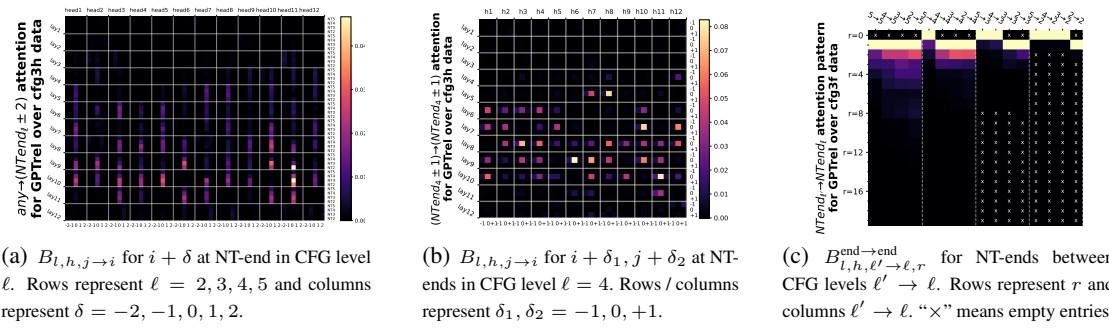

(a) $B_{l,h,j \to i}$ for $i + \delta$ at NT-end in CFG level $\ell$. Rows represent $\ell = 2, 3, 4, 5$ and columns represent $\delta = -2, -1, 0, 1, 2$.

(b) $B_{l,h,j \to i}$ for $i + \delta_1, j + \delta_2$ at NT-ends in CFG level $\ell = 4$. Rows / columns represent $\delta_1, \delta_2 = -1, 0, +1$.

(c) $B^{\text{end} \to \text{end}}_{l,h,\ell' \to \ell,r}$ for NT-ends between CFG levels $\ell' \to \ell$. Rows represent $r$ and columns $\ell' \to \ell$. "×" means empty entries.

Figure 6: Attention has a strong bias towards " NT-end at level $\ell'$ to the most adjacent NT-end at $\ell$ ", for even different $\ell, \ell'$. For definitions see Section 5.2, and more experiments see Appendix E.2, E.3 and E.4.

Our approach differs in that we use synthetic data to demonstrate that linear probing can *almost perfectly* recover NT ancestors and boundaries, even for complex CFGs that generate strings exceeding hundreds of tokens. We focus on pre-training *generative* language models. For a non-generative, BERT-like model pre-trained via language-modeling (MLM), such as the contemporary variant De-BERTa (He et al., 2020), learning *deep* NT information (i.e., close to the CFG root) is less effective, as shown in Figure 4. This is expected, as the MLM task may only require the transformer to learn NT rules for, say, 20 neighboring tokens. Crucially, BERT-like models do *not* store deep NT information at the NT boundaries (see Figure 5).

Our results, along with Section 5, provide evidence that generative language models like GPT-2 employ a dynamic-programming-like approach to generate CFGs, while encoder-based models, typically trained via MLM, struggle to learn more complex/deeper CFGs.

## 5 HOW DO TRANSFORMERS LEARN NTS?

We now delve into the attention patterns. We demonstrate that these patterns mirror the CFG's syntactic structure and rules, with the transformer employing different attention heads to learn NTs at different CFG levels.

### 5.1 POSITION-BASED ATTENTION

We first note that the transformer's attention weights are primarily influenced by the tokens' relative distance. This holds true even when *trained on the CFG data* with *absolute positional embedding*. This implies that the transformer learns the CFG's regularity and periodicity through positional information, which it then uses for generation. (We defer the figures to Appendix E.1 as this finding may not surprise some readers.)

Motivated by this, we explore whether position-based attention *alone* can learn CFGs. In Figure 3, we find that GPT$_{\text{pos}}$ (or even GPT$_{\text{uni}}$) performs well, surpassing the vanilla GPT, but not reaching the full potential of GPT$_{\text{rel}}$. This supports the superior practical performance of relative-position based transformer variants (such as GPT$_{\text{rel}}$, GPT$_{\text{rot}}$, DeBERTa) over their base models (GPT or BERT). **On this other hand, this also indicates that position attention along is not enough for transformers to learn CFGs.**

## 5.2 BOUNDARY-BASED ATTENTION

Next, we *remove* the position-bias from the attention matrix to examine the remaining part. We find that the transformer also learns a strong boundary-based attention pattern, where tokens on the NT-end boundaries typically **attend to the "most adjacent" NT-end boundaries**, similar to standard dynamic programming for parsing CFGs (see Figure 1). This attention pattern enables the transformer to effectively learn the hierarchical and recursive structure of the CFG, and generate output tokens based on the NT symbols and rules.

Formally, let $A_{l,h,j \to i}(x)$ for $j \geq i$ denote the attention weight for positions $j \to i$ at layer $l$ and head $h$ of the transformer, on input sequence $x$. Given a sample pool $\{x^{(n)}\}_{n \in [N]} \in L(\mathcal{G})$, we compute for each layer $l$, head $h$,[6]

$$\overline{A}_{l,h,p} = Average[\![A_{l,h,j \to i}(x^{(n)}) \mid n \in N, 1 \leq i \leq j \leq \mathbf{len}(x^{(n)}) \text{ s.t. } j - i = p]\!] \ ,$$

which represents the average attention between any token pairs of distance $p$ over the sample pool. To remove position-bias, we focus on $B_{l,h,j \to i}(x) := A_{l,h,j \to i}(x) - \overline{A}_{l,h,j-i}$ in this subsection. Our observation can be broken down into three steps.

- Firstly, $B_{l,h,j \to i}(x)$ exhibits a strong bias towards *tokens $i$ at NT ends*. As shown in Figure 6(a), we present the average value of $B_{l,h,j \to i}(x)$ over data $x$ and pairs $i, j$ where $i + \delta$ is the deepest NT-end at level $\ell$ (symbolically, $\mathfrak{b}^\sharp(i + \delta) = \ell$). The attention weights are highest when $\delta = 0$ and decrease rapidly for surrounding tokens.
- Secondly, $B_{l,h,j \to i}(x)$ also favors pairs $i, j$ *both at NT ends* at some level $\ell$. In Figure 6(b), we show the average value of $B_{l,h,j \to i}(x)$ over data $x$ and pairs $i, j$ where $\mathfrak{b}_\ell(i + \delta_1) = \mathfrak{b}_\ell(j + \delta_2) = 1$ for $\delta_1, \delta_2 \in \{-1, 0, 1\}$.
- Thirdly, $B_{l,h,j \to i}(x)$ favors *"adjacent" NT-end token pairs $i, j$*. We define "adjacency" as follows: We introduce $B_{l,h,\ell' \to \ell,r}^{\text{end} \to \text{end}}$ to represent the average value of $B_{l,h,j \to i}(x)$ over samples $x$ and token pairs $i, j$ that are at the deepest NT-ends on levels $\ell, \ell'$ respectively (symbolically, $\mathfrak{b}^\sharp(i) = \ell \wedge \mathfrak{b}^\sharp(j) = \ell'$), and are at a distance $r$ based on the ancestor indices at level $\ell$ (symbolically, $\mathfrak{p}_\ell(j) - \mathfrak{p}_\ell(i) = r$). In Figure 6(c), we observe that $B_{l,h,\ell' \to \ell,r}^{\text{end} \to \text{end}}$ decreases as $r$ increases, and is highest when $r = 0$ (or $r = 1$ for pairs $\ell' \to \ell$ without an $r = 0$ entry).[7]

In conclusion, tokens corresponding to NT-ends at level $\ell'$ statistically have higher attention weights to their *most adjacent* NT-ends at every level $\ell$, *even after removing position-bias*.[8]

**Connection to DP.** Recall that dynamic programming (DP) comprises two components: *storage* and *recurrent formula*. While it's impractical to identify a specific DP implementation that the transformer follows since there are countless many ways to implement a DP, we can highlight *common elements* in DP implementations and their correlation with the transformer. In Section 4, we demonstrated that the generative transformer can encode the DP's *storage* "backbone", encompassing all necessary $\text{DP}(i, j, a)$ on the correct CFG parsing tree of a given string.

For the *recurrent formula*, consider a CFG rule $a \mapsto b, c, d$ in the correct CFG parsing tree. If non-terminal (NT) $b$ spans positions 21-30, $c$ spans 31-40, and $d$ spans 41-50, the DP must establish "memory links" between positions 30-40 and 40-50. This can be achieved by storing the $[bc]$ information at position 40 and merging it with $[d]$ at position 50, or by storing $[cd]$ at position 50 and merging it with $[b]$ at position 30. Regardless of the method, a common feature is the memory link from 30 to 40 and from 40 to 50. Hence, we have been examining such NT-end to NT-end attention links among adjacent NTs in this section.

The transformer is not only a parsing algorithm but also a generative one. Suppose $a \mapsto b, c$ and $c \mapsto d, e, f$ are on the correct parsing tree. When generating symbol $e$, the model, not having finished reading $def$, must access the precomputed knowledge from the uncle node $b$. This is why we also visualized those attentions from an NT-end to its most adjacent NT-end at a different level.

---

[6]Throughout this paper, we use $[\![\cdot]\!]$ to denote multi-sets that allow multiplicity, such as $[\![1, 2, 2, 3]\!]$. This allows us to conveniently talk about its set average.

[7]For any token pair $j \to i$ with $\ell = \mathfrak{b}^\sharp(i) \geq \mathfrak{b}^\sharp(j) = \ell'$ — meaning $i$ is at an NT-end closer to the root than $j$ — it satisfies $\mathfrak{p}_\ell(j) - \mathfrak{p}_\ell(i) \geq 1$ so their distance $r$ is strictly positive.

[8]Without removing position-bias, such a statement might be meaningless as the position-bias may favor "adjacent" anything, including NT-end pairs.

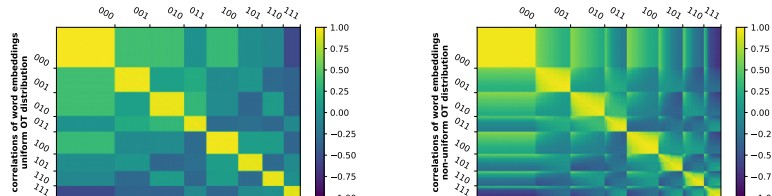

Figure 7: Language models learn implicit CFGs by using word embeddings to encode terminal symbol.

In implicit CFGs, the terminal symbols $t \in \mathbf{T}$ are associated with bags of tokens $\mathbf{OT}_t$ from which observable tokens are sampled. We present word embedding correlations pre-trained on an implicit CFG with $|\mathbf{T}| = 3$ and vocabulary size 300. Details are in Section A.1.

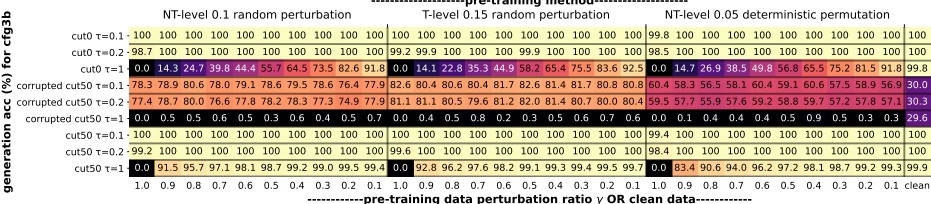

Figure 8: Generation accuracies for models pre-trained cleanly VS pre-trained over perturbed data, on clean or corrupted prefixes with cuts $c = 0$ or $c = 50$, using generation temperatures $\tau = 0.1, 0.2, 1.0$.

**Observation.** In Rows 4/5, by comparing against the last column, we see it is *beneficial* to include low-quality data (e.g. grammar mistakes) during pre-training. The amount of low-quality data could be little ($\gamma = 0.1$ fraction) or large (*every training sentence may have grammar mistake*). The transformer also learns a "mode switch" between the "correct mode" or not; details in Section A.2.

In sum, while defining a good backbone for the DP recurrent formula may be challenging, we have demonstrated several attention patterns in this section that largely mimic dynamic programming regardless of the DP implementations.

## 6 CONCLUSION

**Extensions.** We defer *implicit CFGs* and *robust CFGs* to Appendix A, but briefly showcase the main discoveries in Figure 7 and 8.

**Other related works.** Numerous studies aim to uncover the inner workings of pretrained transformers. Some have observed attention heads that pair closing brackets with open ones, as noted in a concurrent study Zhang et al. (2023). Some have investigated induction heads applying logic operations to the input Olsson et al. (2022). Wang et al. (2022) explored many different types of attention heads, including "copy head" and "name mover head". While our paper differs from these studies due to the distinct tasks we examine, we highlight that CFG is a *deep, recursive* task. Nevertheless, we still manage to reveal that the inner layers execute attentions in a complex, recursive, dynamic-programming-like manner, not immediately evident at the input level.

On the other hand, some studies can precisely determine each neuron's function after training, typically on a simpler task using simpler architecture. For instance, Nanda et al. (2023) examined 1- or 2-layer transformers with a context length of 3 for the arithmetic addition. Our analysis focuses on the inner workings of GPT2-small, which has 12 layers and a context length exceeding 300. While we cannot precisely determine each neuron's function, we have identified the roles of some heads and some hidden states, which correlate with dynamic programming.

**Conclusion.** In this paper, we studied how a transformer learns the CFGs structures in pretraining. CFGs in a language can include grammar, format, expressions, patterns, etc. We consider a synthetic, yet quite challenging family of CFGs to show how the inner workings of trained language models on these CFGs are highly correlated with the internal states of dynamic programming algorithms to parse those CFGs.

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
