# OpenReview forum: "How Language Models Learn Context-Free Grammars"
_ICLR.cc/2024/Conference — ICLR 2024 Conference Withdrawn Submission_

### Official Review · Reviewer_ym3T · 2023-10-30

**Soundness:** 3 good
**Presentation:** 3 good
**Contribution:** 3 good
**Rating:** 6
**Confidence:** 5

**Summary:**

This paper makes two contributions:

1. They investigate the ability of transformer LMs to learn a certain kind of context-free languages (cfg3). They evaluate the success of the models in several ways encompassing both accuracy and diversity (concretely, next-token accuracy, generation diversity, lower bound on support set, and distributional comparison). Across the board, it seems the transformers are capable of learning the cfg3 languages.

2. Mechanistic analysis of how the models are learning to generate the languages using probing.

**Strengths:**

1. The paper is clearly written and two research questions are clearly articulated.
2. The evaluation of the success of the models to learn cfg3 languages is very thorough across different axes (accuracy, diversity, etc.). However, I do have some concerns about the generality of these results for other CFGs (see Weaknesses), which the authors seem to be hinting at in the abstract.

**Weaknesses:**

## More Detail Needed About cfg3

The paper uses a family called cfg3 to benchmark transformers, but surprisingly little is said about this family in the main paper. Even if many of the details are relegated to the appendix, it is crucial to say in the main paper what this particular family of CFG is at a high level and why it was chosen, especially because there are many other more canonical CFLs that have been investigated in previous work such as Dyck languages ([Ebrahimi et al.](https://aclanthology.org/2020.findings-emnlp.384/)) or hardest CFG ([DuSell](https://arxiv.org/pdf/2304.12955.pdf)).

The authors should explain in text or ideally even with a figure example what cfg3 looks like and what the variants are. They should also ideally say something about why they chose cfg3 and how cfg3 compares to more standard CFLs like Dyck languages (in terms of the types of strings it generates, complexity, etc.).

## Unclear What the Authors Mean by "Challenging" CFGs

The authors remark that some of their CFG's are "quite challenging" in the abstract, and later state that the different CFGs within their cfg3 family are ranked by increasing difficulty. Yet, it is not clear what notion of difficulty they mean (this is especially problematic given the lack of detail about cfg3 mentioned above). Presumably, the authors are hinting that these languages are challenging to suggest that the results should be representative for the ability of transformers to learn other CFGs as well. But there is no formal sense presented in which these CFGs are hard or any guarantee or argument about why we should expect results on these languages to transfer to other CFGs.

I would like the authors to either expand on why the languages they chose are challenging or remove any informal claims about them being challenging. Additionally, they should qualify the abstract the indicate that they are studying a particular class of CFGs (cfg3) or else explain why they think the results will translate to other CFGs.

Regarding this point, note that there are canonical examples of hard CFGs like the Dyck languages or "hardest CFG" mentioned above. One other way to address this concern would be to run experiments with these languages as well where there is more of a reason to think the languages in question are "challenging".

## Issues in Probing Results Presentation

Figure 4 should be made larger so it's easier to see the probing results.

Moreover, the results presented in Figure 5 do not fully support the claim in 4.2 that NT ancestor information is locally encoded at NT boundaries and not elsewhere. This is because the plot only shows the results when probing from NT boundaries; I would like to also see a similar plot from other positions in the string where the probing accuracy is much lower.

### nit: Claim about Probing

The authors claim the following about the attention patterns in a transformer:

> If it does encode the NT information after pretraining, it means that the model can both generate and certify sentences in the CFG language

This statement is a bit ambiguous, but one reading is that if a model correctly encodes NT boundaries, it will necessarily be able to generate and certify the CFG. This is actually a fallacy: it has been pointed out many times in the NLP literature that we cannot conclude a model is *using* some grammatical feature correctly just because it is representing it correctly (cf. [Elazar et al.](https://transacl.org/ojs/index.php/tacl/article/view/2423)). In the case of formal language tasks, it may be more unlikely that a model represents a feature it does not use, but in principle the same fallacy still applies. I thus would like the authors to remove or modify this sentence.

**Questions:**

"We use transformer" -> "We use the transformer"

> Moreover, to probe the inner workings of the transformer, we choose a CFG family with a “canonical representation” and show a high correlation between this representation and the hidden states in the learned transformer

What do you mean by a canonical representation? That the CFG is unambiguous or can be recognized left-to-right by a deterministic pushdown automaton?

[Ebrahimi et al.](https://aclanthology.org/2020.findings-emnlp.384/) find that the presence of a beginning-of-sequence symbol is important for learning Dyck languages. Did your models have BOS tokens?

## Missing Related Work

* https://aclanthology.org/2020.findings-emnlp.384.pdf
* https://aclanthology.org/2020.emnlp-main.156/

---

### Official Review · Reviewer_19zR · 2023-10-31

**Soundness:** 3 good
**Presentation:** 3 good
**Contribution:** 3 good
**Rating:** 6
**Confidence:** 3

**Summary:**

The authors study the learning of syntactic structures from the representations of language models (LM). The study uses a probing method based on synthetic data for LMs. The main contributions are: i) probing method of context free grammars (CFG) in LM, ii) visualisation of attention representations and the relation to CFGs, and iii) relation between LMs and dynamic programming (DP). The study shows that LMs encode syntactic structure and learn a relation with DP algorithms used for CFG parsing.

**Strengths:**

- The study proposes a probing method for learning CFG in LMs.
- Clear description of background knowledge and related work needed to understand the proposed study.
- The authors perform a  comprehensive comparison and visualisation of the proposed CFG probing.

**Weaknesses:**

- It is not clearly discussed the hyperparameter selection of the probing and LMs for the study.
- It is not clearly described the implication of using synthetic data on the findings.
- An extra contribution could be to present a statistical significance test or uncertainty estimates of the results.

**Questions:**

Please address the following questions during the rebuttal:

- Please elaborate on the hyperparameter selection for probing (cut and temperature), and LMs.
- Please speculate on the use of natural language with the proposed study, how well the probing method findings will adapt to language?
- Are the findings and/or study can be extended to current GPT models or open-source LLM?

Extra:

Please add previous work on n-gram and/or neural LMs in the introduction for LM in addition to openai.

**Details Of Ethics Concerns:**

I have no concerns.

---

### Official Review · Reviewer_LrUe · 2023-11-03

**Soundness:** 3 good
**Presentation:** 2 fair
**Contribution:** 3 good
**Rating:** 5
**Confidence:** 3

**Summary:**

This paper trains language models (and some masked language models) on a wide variety of strings generated from various context-free grammars.  They then probe the models and find that non-terminal boundaries are well-represented and that patterns in the attention heads correspond to dynamic programming style parsing algorithms for CFGs.  These results show that Transformers do have the ability to learn a wide range of complex CFGs and that they may do so in ways that are natural / interpretable.

**Strengths:**

- Explores formal language learning via language modeling using a very large range of context-free grammars.
- Very extensive set of experiments.
- Probes together with attention patterns begin to provide an understanding of the mechanism that the models have learned.

**Weaknesses:**

- The definition of CFGs and generation therefrom, using layers of non-terminals and whatnot, are somewhat nonstandard.  This makes it a bit hard to follow and digest (e.g. why some grammars are more complex than others).  There also was a related under-description of the grammars that were actually used in the experiments and how, e.g., they relate to more commonly known classes of CFGs like dyck-n.
- There are so many experiments and details that the results and their implications are often under-described (e.g. just "As figure X shows", with a pointer to a complex figure).  This makes it hard for the reader to follow, and easy to lose the forest for the trees.

**Questions:**

- Some of your CFGs are described as having "likely over NUM" strings, for very large numbers.  But all finite languages are in fact regular.  Do you have evidence that the grammars generated have recursive rules, and so actually generate an infinite number of strings in principle?

- Can you say more about the motivation for the multi-linear probe?  In particular, why not just use a linear probe on the last symbol, or use some other simple aggregator of token representations (max or mean pooling)?

---

### Official Review · Reviewer_FLFV · 2023-11-07

**Soundness:** 2 fair
**Presentation:** 1 poor
**Contribution:** 2 fair
**Rating:** 3
**Confidence:** 5

**Summary:**

This paper considers the task of using a Transformer language model to generate suffixes of fixed-depth context-free languages (CFLs). The paper conducts experiments showing that Transformers can learn to do this with a high accuracy of suffix completion and with a reasonable level of diversity. The paper presents analyses based on probing and attention score visualization to demonstrate that the models encode the parse trees of the strings they are generating.

**Strengths:**

The task of generating suffixes for CFLs is an interesting way to think about grammar induction by language models. I like that the paper includes model analysis designed to show that the model really does learn the grammar, as opposed to a mere _imitation_ of the CFL.

**Weaknesses:**

### **Terminology**
There are some imprecise or inaccurate uses of terminology. For example:
- "GPT" is used to refer to the GPT-2 _architecture_ (which is randomly initialized in the experiments), when typically this term is supposed to refer to the pre-trained models. (The GPT-2 architecture is just a typical Transformer decoder.)
- "Dynamic Programming" seems to just refer to any method for CFG parsing that involves keeping track of nonterminal symbols and boundaries. By that logic it's unclear what parsing algorithm would _not_ count as dynamic programming, since it's hard to imagine how to parse without that information.

### **Intuitions**
There are a lot of intuitive claims here that are not justified. For example:
- Under "The cfg3 synthetic CFG family" (p. 3), the text refers to "hard" and "easy" datasets without saying what makes them easy or hard. There is a sentence saying "Typically, the learning difficulty of CFGs _inversely scales_ with the number of NT/T symbols" with a reference to Figure 3, but Figure 3 contradicts this by saying, "Less ambiguous CFGs (... as **they have fewer NT/T symbols** [emphasis added]) are **easier to learn** [emphasis added]."
- The "Why Such CFGs" section (p. 3) says, "The probability of a random string belonging to this language is nearly zero," but the paper presents no evidence, citation, or theoretical justification for this.
- The paper refers to "highly ambiguous" and "less ambiguous" CFGs without saying how they measure ambiguity. The Figure 3 caption suggests that "less ambiguous" means "fewer nonterminals," but it's not immediately obvious that this would be the case (e.g., consider a rule like $A \to A$).

### **Methodology**
- The main experiments have no baseline. Although the entropy experiment has a "ground truth" value, it's still not clear how easy or hard the task is.
- I'm somewhat unconvinced by the probing experiment. It seems to me that in theory, the nonterminal label and boundary information should be fully recoverable from the next token distribution alone. Therefore if the language model really does generate competently from the CFL, then probing for this information should be trivial.

### **Presentation and Framing**
- The paper makes many references to the "how" of "learning": for example, the title is _**How**_ [emphasis added] _Language Models **Learn**_ [emphasis added] _Context-Free Grammars_. However, the paper never actually addresses the question of _how_ the models learn; it only analyzes models _after_ learning is complete.
- The paper is overly reliant on the appendix. The appendix should be reserved for implementation details and proofs of mathematical claims, and it shouldn't be essential to a typical reader's understanding of the paper.
- Some of the notations and definitions are overly cumbersome. For example, the description of the generation algorithm and the notation for boundaries could be greatly simplified.

**Questions:**

- Why use fixed-depth CFGs?
- Why use binary and ternary rules instead of Chomsky normal form?

---

### Public Comment · ~Ada_Wan1 · 2023-11-22
**Please no more grammar in LMing!**

The premise of this paper is questionable. The paper implicitly assumes a relation between language models (LMs) and context-free grammars (CFGs) and that LMs learn CFGs. As most literature in the past also made similar assumptions (and often without rigorously assessing the relevance of grammar at all when it comes to real-world text data), it would certainly not be an unfair question to ask _again_ to which extent grammar has anything to do with LMing, esp. considering recent literature such as [1].

Re "Despite recent theoretical advances in understanding language models ... most are limited to simple settings and fail to account for the complex structure of languages.":
what "complex structure of languages" are there to account for? As both [1] and [2] point out, when it comes to LMing, it is a matter of data statistics from sequence length and vocabulary.

It seems that this paper can be improved through some explicit discussions of data statistics, instead of trying to pull "grammar" into the picture. That would be a sensible continuation of a scientific/scholarly discourse on LMing.

Last but not least, there is an ethical concern when it comes to "grammar" --- it assumes well-formedness and with such, there is some implicit, inappropriately divergent sentiment/direction that is being propagated.

[1] Sabrina J. Mielke, Ryan Cotterell, Kyle Gorman, Brian Roark, and Jason Eisner. What kind of language is hard to language-model? https://arxiv.org/abs/1906.04726.

[2] Ada Wan. Fairness in representation for multilingual NLP: Insights from controlled experiments on conditional language modeling. In International Conference on Learning Representations, 2022. https://openreview.net/forum?id=-llS6TiOew.